# C2D2 Dataset: A Resource for Analyzing Cognitive Distortions and Its Impact on Mental Health

**Bichen Wang, Pengfei Deng, Yanyan Zhao,*** **Bing Qin**
Research Center for Social Computing and Information Retrieval
Harbin Institute of Technology, Heilongjiang, China
{bichenwang,pfdeng,yyzhao,qinb}@ir.hit.edu.cn

## Abstract

Cognitive distortions refer to patterns of irrational thinking that can lead to distorted perceptions of reality and mental health problems in individuals. Despite previous attempts to detect cognitive distortion through language, progress has been slow due to the lack of appropriate data. In this paper, we present the C2D2 dataset, the first expert-supervised **C**hinese **C**ognitive **D**istortion **D**ataset, which contains 7,500 cognitive distortion thoughts in everyday life scenes. Additionally, we examine the presence of cognitive distortions in social media texts shared by individuals diagnosed with mental disorders, providing insights into the association between cognitive distortions and mental health conditions. We propose that incorporating information about users' cognitive distortions can enhance the performance of existing models mental disorder detection. We contribute to a better understanding of how cognitive distortions appear in individuals' language and their impact on mental health.

## 1 Introduction

Cognitive distortions are irrational thinking patterns that can lead to distorted perceptions of reality (Beck, 1970). For example, the thought *"I lost my puppy, and my future will be perpetually filled with sadness and loneliness"* exemplifies a cognitive distortion known as "Overgeneralization."In this case, the conclusion implies an excessively broad and permanent state of unhappiness. A more precise and less distorted thought for this situation would be: *"The sudden loss of my puppy was incredibly painful, and I struggled to come to terms with it."*

As depicted in Figure 1, cognitive distortions significantly hinder individuals' perception, trapping them further through continuous self-reinforcement and contributing to the development of mental disorders, including depression, anxiety, and post-

---
* Corresponding author

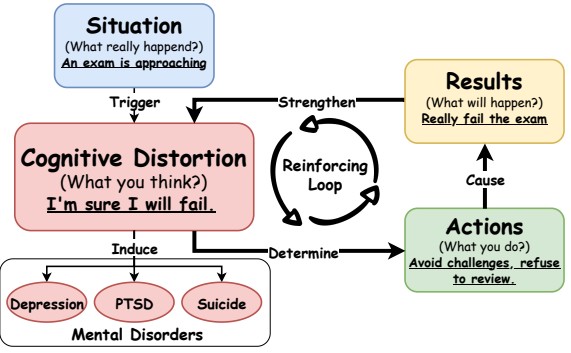

Figure 1: Cognitive distortions' impact on individuals. Cognitive distortions are constantly strengthened within this reinforcing loop, contributing to the development of mental disorders such as depression, PTSD, and anxiety (Burns, 1981).

traumatic stress disorder (PTSD) (Marton et al., 1993; Muris and Field, 2008; Abel et al., 1989; Strohmeier et al., 2016; Hammen, 1978). Hence, the development of automated tools for detecting cognitive distortions is important, as it assists researchers in early detection cognitive distortions, facilitating timely intervention strategies aimed at enhancing individuals' psychological well-being and happiness.

Previous research demonstrates that computational techniques can effectively detect cognitive distortions from language (Shickel et al., 2020; Simms et al., 2017; Bathina et al., 2021). However, these studies have been conducted using private datasets or datasets with limited and low-quality annotations (Alhaj et al., 2022; Ziems et al., 2022). Despite cognitive psychology acknowledging an association between mental disorders and cognitive distortions, the absence of a reliable and publicly accessible dataset has obstructed the establishment of a credible benchmark and further advancements in this field.

To tackle these challenges, we introduce our C2D2 dataset, a Chinese dataset created to address

the shortage of research resources regarding cognitive distortions. It is a publicly accessible resource within this domain that includes 7,500 instances of cognitive distortion thoughts from 450 different scenes. The research of our C2D2 can promote research on cognitive distortion and provide insights into users' mental state. We believe this resource will contribute to mental health research in China, a developing country, particularly in light of increasing social pressure and inadequate support for the mental healthcare system.

Moreover, we conduct various experiments, including the cognitive distortion detection and exploring the relationship between cognitive distortions and mental disorders. We compare the performance of finetuning different pretrained models and large language models using in-context learning on C2D2 dataset. We demonstrate the current models' ability to detect cognitive distortions. Furthermore, our research is not limited to cognitive distortions alone. Inspired by psychology, we use computational methods to investigate the association between cognitive distortions and mental health. We use several datasets on mental health to detect and analyze present cognitive distortions in the online texts of individuals with different disorders using our model. We discover some interesting phenomena and conclusions that have not been previously considered, but could potentially validate the underlying mechanisms of certain disorders. Finally, we develop a simple method to enhance the performance of detecting disorders based on cognitive distortions. The creation of the C2D2 dataset contributes to a better understanding of cognitive distortions and the impact of cognitive distortions on mental health.

- We have developed a publicly accessible Chinese Cognitive Distortion Dataset[1] for the first time, aiming to facilitate the analysis of cognitive distortions and promote the integration of psychology and computational technology.

- We explore the association between cognitive distortions and various mental disorders on social media platforms. To our knowledge, this is the first work that utilizes cognitive distortion to assess users' mental states.

- We attempt to incorporate information about

users' cognitive distortions into mental disorder detection models, emphasizing the significance of including cognitive distortion computations.

## 2 Related Work

In the following sections, we review studies pertaining to cognitive distortions, covering both psychological research and computational techniques.

### 2.1 Cognitive Distortion in Psychology

Since the emergence of cognitive-behavioral theory, cognitive distortions have been the subject of extensive research (Beck, 1970). This line of study, enriched by various researchers, has cultivated a comprehensive theoretical framework (Beck, 2020, 1979; Burns, 1981).

Although originating in depression research, cognitive distortions have also been linked to a variety of issues such as pathological gambling, anxiety, suicide, and anorexia (Marton et al., 1993; Muris and Field, 2008; Abel et al., 1989; Strohmeier et al., 2016; Fortune and Goodie, 2012). From a sociological perspective, they are hypothesized to correlate with juvenile delinquency and antisocial personality (Nas et al., 2005; Wallinius et al., 2011; Gannon and Polaschek, 2006; Feldman, 2007). Furthermore, another branch of research aims to guide psychologists in detecting and rectifying these distortions, thereby studying their impact on treatment (McClenahan, 2005; Yurica and DiTomasso, 2005).

### 2.2 Cognitive Distortion Detection and Application

In the study of mental health computing, the main focus includes emphasis on emotion analysis and symptom identification (Gkotsis et al., 2016; Shickel et al., 2016). Some researchers conduct research from the perspective of cognition and emotion (Uban et al., 2021). However, the application of machine learning methods from a cognitive-behavioral perspective, however, has garnered less attention. Prior work has explored the use of machine learning to detection cognitive distortions in mental health texts or on social media (Shickel et al., 2020; Simms et al., 2017; Wang et al., 2022), as well as within medical dialogues between physicians and patients (Shreevastava and Foltz, 2021; Tauscher et al., 2023).

The limited exploration in this field can be attributed to the lack of publicly available, well-

[1]Our dataset will be available in https://github.com/bcwangavailable/C2D2-Cognitive-Distortion.git

| Cognitive Distortion | Definition | Example |
|---|---|---|
| Black and white thinking | Oversimplification of complex situations into binary categories. | "If I don't get an A on this test, I have completely failed." |
| Emotional reasoning | Belief that what we feel must be true, regardless of the evidence. | "I feel like a bad, so I must be." |
| Fortune-telling | Predicting negative outcomes without basis. | "I know I will do poorly in the presentation tomorrow." |
| Labeling | Attaching arbitrary labels to oneself or others. | "I made a mistake, I am a loser." |
| Mindreading | Belief of accurately discerning others' thinkings without direct communication. | "I can tell they're laughing at me behind my back." |
| Overgeneralization | Making sweeping negative conclusions based on limited instances. | "I failed one exam, I'm going to fail the whole course." |
| Personalization | Attributing the negative behaviors of others to oneself without plausible connections. | "My boss is upset, it must be something I did wrong." |

Table 1: Definitions and Examples of Cognitive Distortions. Our data contains seven common cognitive distortions.

annotated datasets. Previous studies on computational mental health primarily focus on emotions and symptoms. However, we believe that the impact of cognitive distortions and thinking patterns on mental health is also significant.

## 3 Dataset Construction

Previous researchers have collected posts from social media platforms or gathered data related to cognitive distortions through crowdsourced writing (Shickel et al., 2016; Alhaj et al., 2022). We have chosen not to directly annotate social media content due to the inability to effectively control its quality and accuracy. Instead, we have adopted a specially designed task to collect cognitive distortion thoughts. Unlike previous work that recruit volunteers widely from the internet, our data annotation process is executed through a collaborative effort between carefully selected and specifically trained volunteers and domain experts. As depicted in Figure 2, we will further describe our task design and the three phases, which include volunteer recruitment and screening, data annotation, and expert evaluation.

### 3.1 Data Annotation Target

Psychologists have identified various categories of cognitive distortions that are often exhibited in individuals' thought (Beck, 1970, 1979; Ellis, 1994). In our work, the C2D2 dataset encompasses seven typical cognitive distortions, which are shown in Table 1. Our annotation task involves providing volunteers with scenes and requesting them to write down multiple possible cognitive dis-

tortion thoughts based on the given scenes. An example is presented in Table 2, it simply displays the translated content of our dataset. It should be noted that different types of cognitive distortions do not strictly appear independently. However, for the purpose of simplifying the annotation process, we have treated it as a single-label task. The volunteers' goal is to generate instances that represent a single type of distortion. In situations where multiple cognitive distortions occur simultaneously, we ask volunteers to select the label of the dominant cognitive distortion.

---

**We provide the following to volunteers:**

*Scene*: The company's project is about to be delivered, and you caught a cold at this critical juncture

*Scene category*: Work issues

---

**The content completed by volunteers is as follows:**

*Cognitive Distortion*: Cold & overtime, bad luck peaks! Why all misfortune on me!(Overgeneralization)

*Cognitive Distortion*: Is my illness at such a critical moment a sign that something will go wrong with the project as well? (Fortune-telling)

*Non-distorted*: Feeling terrible with a cold - my nose and throat are both sore. So painful!(Non-distorted)

. . . At least 5 different types of cognitive distortions and 2 normal thought.

---

Table 2: An instance from C2D2 dataset. This example is translated into English.

### 3.2 Volunteer Recruitment & Screening

**Psychology Questionnaire** We recruit volunteers to participate in our data collection on cognitive distortions. They complete a cognitive distortion questionnaire (Covin et al., 2011). We select volunteers

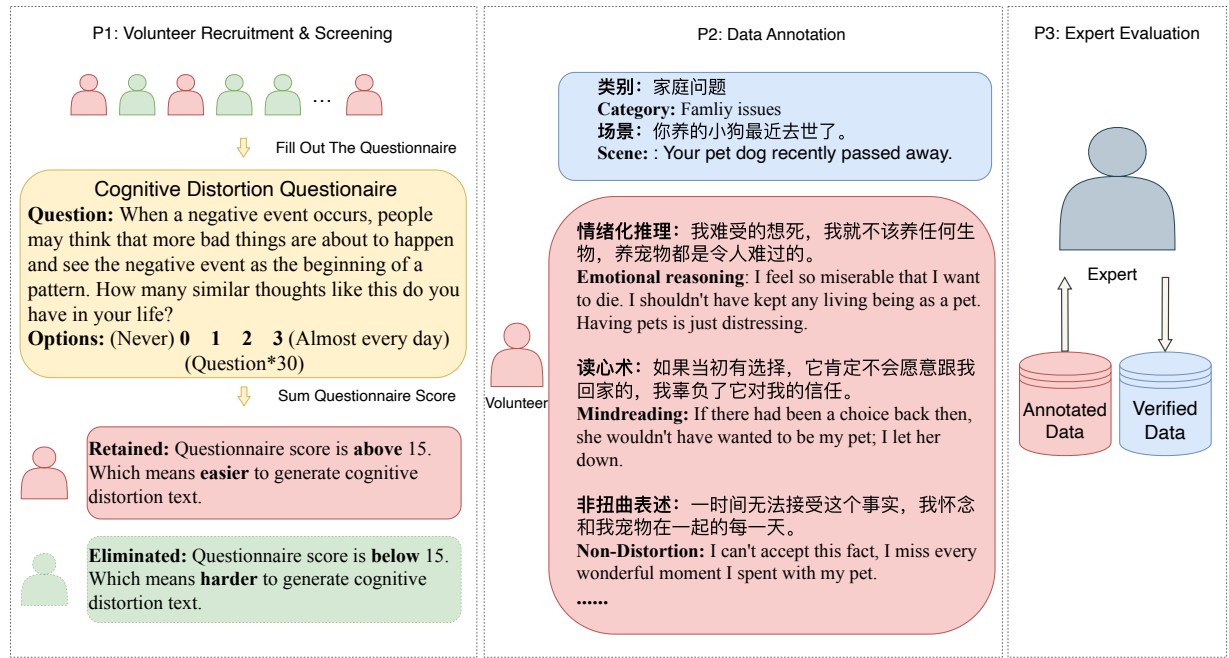

Figure 2: The Data Collection Process includes volunteer recruitment&screening, data annotation, and conducting a final expert evaluation. During the volunteer recruitment phase, we carefully select volunteers using a cognitive distortion questionnaire. The data annotation collect cognitive distortions. The expert evaluation guaranteed the reliability of our data.

based on their questionnaire results, prioritizing those who show a potential for exhibiting cognitive distortions in daily life. This strategy enables us to collect more authentic cognitive distortion thoughts.

**Volunteer Training** To ensure our volunteers understand cognitive distortions and can recognize them in their daily lives, we provide a comprehensive psychology training program. This program includes an examination on the identification and understanding of common cognitive distortions. After two days of cognitive psychology training and an introduction to the harms of cognitive distortions, we retain 24 out of the initial 50 volunteers. These volunteers demonstrate an ability to identify cognitive distortions and acknowledge their harmful effects. All volunteers agree to our data collection and public disclosure requirements. [2]

### 3.3 Data Annotation

**Scene Preparation** As depicted in Figure 2, we prepare 7 categories of 450 daily scenes, including *work issues* (20%), *interpersonal issues* (20%), *economic issues* (10%), *random negative events* (5%), *family issues* (25%), *physical stress* (10%), and *dis-*

*crepancy between ideal and reality* (10%) (Lazarus and Folkman, 1984). These categories cover various aspects of life that can trigger cognitive distortions. By providing these scenes, we encourage the volunteers to imagine themselves in these situations and describe how they would perceive the events from the perspective of someone experiencing cognitive distortion thoughts.

**Data Collection** Volunteers are asked to adopt the perspective of an individual experiencing cognitive distortions and generate thoughts related to a specific cognitive distortion that may arise in the given scene. Moreover, volunteers are asked to label thoughts provided by other volunteers. The final cognitive distortion labels for our data are determined by majority vote among three volunteers, reducing the influence of individual bias. Our primary objective is cognitive distortion detection; therefore, volunteers are assigned overlapping events for data collection.

### 3.4 Expert Evaluation

Our annotated data undergo rigorous expert evaluate to ensure quality. Experts randomly sample from each volunteer-contributed dataset, evaluating each volunteer's work comprehensively. If the data do not meet the standards, volunteers receive feed-

---

[2]For the protection of volunteers' rights, please refer to Section 9.1 and Appendix A.

| Evaluation Criteria | Dataset (Shickel et al., 2020; Simms et al., 2017; Shreevastava and Foltz, 2021) | | | |
| --- | --- | --- | --- | --- |
| | C2D2(Our Dataset) | MH | Tumblr Dataset | Therapist |
| Public availability | ✓ | ✗ | ✗ | ✓ |
| Multiple cognitive distortion types | ✓ | ✓ | ✗ | ✓ |
| Training of annotators | ✓ | ✗ | ✗ | ✗ |
| Consistency in annotation | ✓ | ✓ | ✓ | ✗ |
| Scale of dataset | 7,500 | 2,769 | 1,728 | 3,136 |

Table 3: Comparison of the key features of our work with previous studies.

back, are asked to revise their submissions, and may receive additional training. The evaluation criteria include:

- Correctness: Experts examine if the assigned labels accurately indicate cognitive distortions.

- Reasonableness: They evaluate if this cognitive distortion aligns with the cognitive distortions that may occur in real-life thinking.

- Emotional Diversity: They verify if the data reflects a broad spectrum of emotions, highlighting the emotional complexity in human cognition.

## 3.5 Data Quality Assurance

To ensure the quality of C2D2, we undertake multiple evaluations. Initially, we compute the inter-annotator $Kappa$ score among our volunteers in regard to our labels. The outcomes reflect a moderate degree of concordance, with a mean $Kappa$ score of 0.67, signifying significant consensus among the annotators. Subsequently, we engage experts to examine our complete dataset. These experts are requested to grade the data on a 1 to 5 scale. The assessments return high scores in all categories, presenting mean ratings of roughly 4.7 for Correctness, 4.5 for Emotional Diversity, and 4.1 for Reasonableness. These elevated scores corroborate that our dataset is well-aligned with the task instructions and embodies the necessary characteristics to train a model proficient in detecting cognitive distortions.

## 4 Data Characteristics

## 4.1 Statistics

The provided dataset statistics are displayed in Table 4, showcasing the overall characteristics of the dataset. It can be seen that the label distribution of our dataset is relatively uniform.

| Category | Num | Avg.Len |
| --- | --- | --- |
| Black and white thinking | 690 | 29.45 |
| Emotional reasoning | 751 | 30.69 |
| Fortune-telling | 682 | 30.91 |
| Labeling | 721 | 29.11 |
| Mindreading | 1,003 | 30.11 |
| Overgeneralization | 894 | 29.58 |
| Personalization | 709 | 31.36 |
| Non-distorted | 2,050 | 28.14 |
| Sum | 7,500 | 29.68 |

Table 4: Dataset Statistics

After the completion of data collection, our dataset comprises a total of 7,500 thoughts. These thoughts are divided into three sets for training, validation, and testing purposes, following an 8:1:1 ratio.

## 4.2 Related Datasets

As shown in Table 3, our C2D2 dataset has many advantages compared to previous works. It is, firstly, a publicly accessible resource within this domain that includes numerous texts recording individuals' thoughts in various scenes. Secondly, we have annotated each thought, not only indicating the presence of cognitive distortion but also categorizing them according to cognitive psychology (Beck, 1979). Finally, we've maintained data authenticity and privacy through comprehensive annotator training, backed by a stringent three-phase data collection process.

The data collection methods of previous works included social media platforms or gathered data related to cognitive distortions through crowdsourced writing. Due to the privacy and sensitivity of mental health data, collaborating with trained volunteers and domain experts to construct reliable synthetic data is a viable alternative. Our approach ensures data quality and authenticity while maximiz-

ing the availability of open datasets for researchers, thereby breaking the barriers of data privacy and low data quality that currently exist.

Our data focuses on the Chinese language, particularly in China, a developing country. Previous researches on mental health predominantly conduct in developed regions with advanced psychological resources. However, mental health issues in developing countries often receive limited attention. In these countries, there is a greater need for affordable and dependable automated detection technologies compared to the availability of reliable mental health services in developed regions (Patel and Kleinman, 2003; Kohn et al., 2004; Chowdhary et al., 2014).

## 5  Cognitive Distortion Detection

We develop a model to detect cognitive distortions in text and assign specific categories to these cognitive distortions. This detection and categorization can facilitate interventions by psychologists and further analysis for treatment purposes. The task, Cognitive Distortion Detection, involves inputting text $X$ containing cognitive distortions and generating a multi-class label $y \in \{0, 7\}$. Here, $0$ denotes non-distorted, while the other values represent distinct categories of cognitive distortions.

### 5.1  Baseline Models and Results

To evaluate the task, we employ pretrained language models on our C2D2 dataset. Table 5 presents the results of the baseline models. Our objective is to provide researchers with a dataset and establish a benchmark. We finetune the different kind of Chinese versions of the pretained language models (Cui et al., 2021, 2020). Additionally, we assess the few-shot and zero-shot settings for LLM, such as ChatGPT (Ouyang et al., 2022), providing three examples for each class in few-shot learning. [3]

The pretrained models demonstrate satisfactory performance. However, LLM's in-context learning performance does not match that of the finetuned models for this particular psychological task. Nevertheless, significant improvement is observed with a few examples in the few-shot learning setting. We have built a cognitive distortion detection model based on various models. We develop cognitive distortion detection model utilizing multiple

---

| Model | $F1$ | $Rec$ | $Pre$ |
|---|---|---|---|
| Bert | 0.69 | 0.73 | 0.68 |
| Roberta | **0.73** | **0.75** | 0.72 |
| XLnet | 0.72 | 0.71 | **0.73** |
| Electra | 0.70 | 0.69 | 0.71 |
| ChatGPT (Zero-shot) | 0.39 | 0.37 | 0.41 |
| ChatGPT (Few-shot) | 0.51 | 0.49 | 0.54 |

Table 5: Performance of baseline models for the C2D2 tasks. All metrics are calculated using macro-averaging.

models. We believe that tasks such as cognitive distortion detection, which demand specialized analytical capabilities and expertise rather than being universally accessible, warrant increased attention in the future.

## 6  Cognitive Distortion for Mental Health

After constructing the cognitive distortion detection task, we aim to introduce the importance of cognitive distortions in mental health. Previous computational research has concentrated on examining the emotions and symptoms exhibited by individuals with mental disorders. In contrast, our objective is to find out the underlying cognitive distortion. This section sims to examine two fundamental questions in order to underscore the significance of cognitive distortion detection.

**Q1: How do thinking patterns differ between individuals diagnosed with mental disorders and the normal group?**

In Section 6.2, we employ our cognitive distortion detection model to analyze social media posts from individuals diagnosed with depression and PTSD, as well as normal group without any self-reported mental health issues. From a cognitive psychology perspective, we aim to identify the cognitive distortions to detect differences between these groups.

**Q2: Can these differences be utilized for mental disorder detection?**

In Section 6.3, we simply integrate the cognitive distortions into the method for detecting mental disorders, with a specific focus on depression and PTSD. This approach underscores the potential utility of cognitive distortion to improve mental disorder detection, particularly when conventional research largely concentrates on symptoms and emotional manifestations.

---

[3]Please refer to Appendix B and Appendix C for more details.

| Dataset | | Depression | PTSD | Normal |
|---|---|---|---|---|
| CLPsych-2015 | Num.subjects | 477 | 396 | 873 |
| | Num.posts | 1,083,403 | 868,428 | 2,266,740 |
| | Avg num. of posts per subject | 2,271.5 | 2,193.4 | 2,596.5 |
| | Avg length.of per posts | 13.91 | 13.52 | 13.88 |
| eRisk-2018 | Num.subjects | 135 | - | 752 |
| | Num.posts | 49,557 | - | 481,837 |
| | Avg num. of posts per subject | 367.1 | - | 640.7 |
| | Avg length.of per posts | 45.13 | - | 34.11 |

Table 6: Mental disorder data statistics

## 6.1 Mental Disorder Dataset

We utilize two different datasets. Table 6 illustrates the datasets that we have used, which focus on PTSD and depression, leveraging data from Twitter and Reddit respectively. The CLPsych-2015 dataset (Coppersmith et al., 2015) consists of Twitter data from normal individuals diagnosed with depression, PTSD, and those normal group without mental disorders. On the other hand, the eRisk-2018 dataset (Losada et al., 2018) incorporates Reddit data from individuals diagnosed with depression and normal group.

The original C2D2 dataset is in Chinese. Given that most existing mental health datasets are in English, we have translated the C2D2 into English, producing the C2D2-E (English version of C2D2). We have used machine translation tools for this purpose and performed sampling inspections to ensure quality. The label distribution and content of C2D2-E mirror those of the original C2D2 dataset.

## 6.2 Cognitive Distortions Analysis on Social Media

In this section, we use the BERT-based baseline model, trained on the C2D2-E dataset, to detect cognitive distortions in social media posts from individuals diagnosed with depression and PTSD, as well as normal group. We focus on determining the prevalence of cognitive distortions on social media platforms. We define cognitive distortion prevalence as $p_{freq}$, which is computed as follows:

$$p_{freq} = \frac{N_{cd}}{N_{nor} + N_{cd}} \quad (1)$$

where $N_{nor}$ represents the number of normal post data, and $N_{cd}$ represents the number of posts containing cognitive distortions. We calculate the $p_{freq}$ for each user and present the results in a box plot.

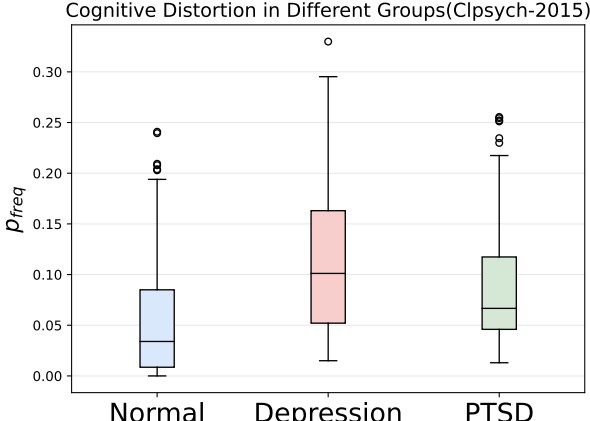

Figure 3: Box plot of the distribution of $p_{freq}$ across different groups. The vertical axis represents the $p_{freq}$ of cognitive distortions, and the horizontal axis represents the mental disorders of the users.

Figure 3 illustrates that the $p_{freq}$ of cognitive distortions expressed on social media by individuals diagnosed with depression is higher than that of both the general population and individuals with PTSD using the CLPsych-2015 dataset. [4] Among individuals diagnosed with depression, PTSD, and the normal group, the average $p_{freq}$ is 13% for depression, 8% for PTSD patients, and 3% for the normal group. ***Our results indicate that cognitive distortions are most prominent in patients with depression, followed by those with PTSD, compared to individuals without any reported mental disorders.*** While cognitive distortions exist among the normal group, they are less prevalent compared to individuals diagnosed with mental disorders. By comparing $p_{freq}$ across various mental disorders, we validate the relationship between cognitive distortions and mental disorders, indicating that an increase in cognitive distortions could potentially

---

[4] eRisk result is available in Appendix D.

serve as an overlooked characteristic for detecting mental disorders.

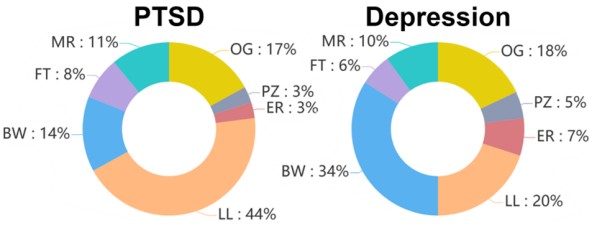

Figure 4: Category Differences in Cognitive Distortions among Different Mental Disorders. MR stands for Mind-Reading, FT stands for Fortune-Telling, BW stands for Black and White thinking, LL stands for Labeling, OG stands for Overgeneralization, PZ stands for Personalization, and ER stands for Emotional Reasoning.

Figure 4 shows our analysis of the average distribution of cognitive distortion types related to PTSD and depression using the CLPsych-2015 dataset. *Our results reveal distinct cognitive distortion patterns across different mental disorders. Specifically, individuals diagnosed with depression show a heightened tendency towards emotional reasoning and black and white thinking. In contrast, patients diagnosed with PTSD display a significant occurrence of labeling.* This unexpected finding suggests that PTSD patients may excessively categorize benign situations as potential threats due to their traumatic experiences (Brewin, 2001; Van der Kolk, 2022). Our study of social media data underscores the importance of investigating cognitive distortions, potentially contributing to the improvement of therapeutic methods for treating PTSD. Moreover, our results highlight the feasibility of using computational techniques to analyze underlying psychological phenomena from data.

### 6.3 Enhancing Mental Disorder Detection via Cognitive Distortion

After discovering the aforementioned phenomenon, we attempted to incorporate cognitive distortion features into a simple mental disorder detection model. Our approach involves analyzing a user's posting history, denoted as $P$, and extracting posts that contain cognitive distortions, denoted as $C$, using a finetuned BERT model. (e.g., "Overgeneralization: I failed an exam, so the whole discussion will fail."). By integrating cognitive distortions into the detection process, we aim to improve the performance of detecting mental disorders.

**Method** Our approach involves utilizing an LSTM to capture the user's historical context and model

cognitive distortions. The resulting hidden states are then combined with post popularity in a feed-forward neural network for final classification. The simple formula is shown as follows:

$$h_p = BiLSTM(Bert(P)) \qquad (2)$$

$$h_c = BiLSTM(Bert(C)) \qquad (3)$$

$$y = FFN([h_c : h_p : p_{freq}]) \qquad (4)$$

where $h_p \in \mathbb{R}^{128}$. We incorporate content features and numerical features related to cognitive distortion in our model. Our experiment was very straightforward, but we aim to demonstrate the role of cognitive distortion through this direct approach.

**Result** We conduct a preliminary evaluation of incorporating cognitive distortion information into the detection of mental disorders. The results show some improvement in detection performance when including users' cognitive distortion information in the model. This approach is effective across different platforms and in detecting various mental disorders.

| Dataset | Model | $F1$ | $Rec$ | $Pre$ |
|---|---|---|---|---|
| CLPsych2015-depression | Bert-LSTM | 0.73 | 0.75 | 0.72 |
| | **+CD feature** | **0.79** | **0.77** | **0.82** |
| CLPsych2015-ptsd | Bert-LSTM | 0.74 | 0.76 | 0.73 |
| | **+CD feature** | **0.85** | **0.87** | **0.84** |
| eRisk2018-depression | Bert-LSTM | 0.59 | 0.62 | 0.59 |
| | **+CD feature** | **0.65** | **0.63** | **0.65** |

Table 7: Comparison of performance metrics for different models on multiple datasets. The method we use and the obtained values are shown in bold.

In various mental disorders across different platforms, the small modifications we made by incorporating cognitive distortions yielded superior results compared to the original model. We believe that our experiment demonstrates the potential to enhance the performance of existing mental disorder detection models through modeling cognitive distortions.

## 7 Conclusion

We have introduced the C2D2 dataset, which is the first public Chinese dataset focused on cognitive distortions. This dataset illuminates a profound connection between cognitive distortions and mental disorders. Addressing these distortions could potentially enhance mental health interventions. By integrating cognitive distortions as an additional

feature, we have augmented the performance of existing mental disorder detection models. The introduction of the C2D2 dataset represents a valuable contribution to computational psychology and serves as a catalyst for further research in this emerging field.

# 8 Limitation

While collecting data, we try our best ensure its quality. However, our data may still have potential biases. Nevertheless, the data's inherent value remains unchanged.

The observed phenomena in mental illness data are validated using two mental health datasets, with the comparison between PTSD and depression based on a single dataset. These datasets may have biases, but gathering diverse user-level data on various mental illnesses is challenging for us. Therefore, we don't claim our findings as definitive in psychology. However, we believe that these phenomena are likely to be widespread. We encourage future researchers to analyze them further with more data. Recognizing and interpreting the data require additional endorsement from psychology experts, beyond the scope of our work.

We believe that there are many areas within our dataset that can be further explored and utilized, and our work only covers a small portion that we consider representative but limited.

# 9 Broader Impact and Ethical Considerations

## 9.1 Ethical Considerations

When dealing with sensitive data such as the psychological well-being of human subjects, special care must be taken. In this case, our main goal is to provide a dataset for the general public, which makes confidentiality even more important.

Our research has received approval from the Institutional Review Board (IRB) of our institution. All data annotators involved are over 18 years old and have signed informed consent forms agreeing to the public release of the data. We have removed any data that may be personally identifiable to the data subjects. Additionally, during the data annotation process, we had a mental health expert monitoring the psychological well-being of the volunteers to ensure the well-being of the annotators. Volunteers had the option to withdraw from the process at any time. Finally, the remaining data we used is from publicly available datasets, and the ethical considerations for these datasets have been guaranteed by the dataset creators.

## 9.2 Positive Outcomes

- Improved Mental Health Services: Therapists and psychologists could use these techniques to identify cognitive distortions quickly, allowing them to dedicate more time to therapeutic processes.

- Early Detection and Intervention: We could enable early detection of mental health disorders, leading to timely interventions. By identifying cognitive distortions on social media platforms, professionals might be able to reach individuals who need help but haven't sought treatment.

- Accessible Mental Health Resources: The public availability of the C2D2 dataset encourages other researchers to continue this important work. Increased research could lead to more accessible mental health resources, further destigmatizing mental health and supporting individuals who might not otherwise have access to care.

## 9.3 Negative Outcomes and Mitigation Strategies

- Misinterpretation of Data: Automated systems can make mistakes, and those mistakes could have serious consequences in the realm of mental health. An incorrect interpretation of a person's statements could lead to unnecessary worry or intervention, or it might miss a person who genuinely needs help.

- Over-reliance on Technology: While our work could significantly aid mental health professionals, there's a risk that some may become over-reliant on these computational techniques. Human judgment and intuition remain essential in mental health services, and a balance between technological and human input must be maintained.

## Acknowledgments

We thank the anonymous reviewers for their insightful comments and suggestions. This work was supported by the National Key RD Program of China via grant 2021YFF0901602 and the National Natural Science Foundation of China (NSFC) via grants 62176078 and 62236004.

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

## A  Ensuring Volunteer Well-being

To prioritize the mental health of our volunteers, we have implemented the following measures throughout the data collection process:

- Precautions: Prior to participation, volunteers are provided with detailed information about the task, including potential exposure to challenging situations and cognitive distortions.

- Informed Consent: Volunteers provide informed consent before engaging in the data collection process. We emphasize that their participation is entirely voluntary, and they have the freedom to withdraw at any time without facing any consequences.

- Supportive Environment: We maintain open channels of communication with volunteers, encouraging them to share any concerns or difficulties they may encounter during the task.

- Anonymity and Confidentiality: Volunteers are assured that their identities will remain anonymous and their personal information will be kept confidential. This fosters a safe space for open and honest participation.

By implementing these measures, we demonstrate our commitment to the well-being of our volunteers and ensure an ethical and responsible data collection process.

## B  Training Details

We mention our testing of the large language model and finetuning of the pretrained model. Specifically, our prompt using in LLM and the training details are as follows.

**Description:** Cognitive distortions refer to detrimental patterns of thinking that are prevalent in individuals. Assistance is required to identify these patterns of thinking based on the textual content. The following are the distinct categories of cognitive distortions along with their precise definitions:
**Definite:** Definitions of seven types of cognitive distortions
**Examples:** A total of twenty-four illustrative instances
**Thought:** Input data

For Cognitive Distortion Detection, the learning rate is set to 1e-5, and the AdamW optimizer is employed. The base version of the model, without any modifications, is used. To determine the best performing model, the learning rate and optimizer are chosen based on empirical observations and previous studies in the field. The model exhibiting the lowest loss on the validation set, which is separated from the training set, is selected for testing. The results are averaged over 5 runs to ensure robustness and mitigate the effects of random initialization.

For Mental Disorder Detection, both non-pretrained models and the pretrained model, in their base versions, are utilized. The learning rate for non-pretrained models is set to 1e-3, while the pretrained model has a learning rate of 1e-6. The LSTM hidden layer state is configured with 128 dimensions to capture complex temporal dependencies. Similar to the Cognitive Distortion Detection, the AdamW optimizer is employed. The official test set is used for evaluation, and a separate validation set is created from the training set. The model exhibiting the lowest loss on the validation set is selected for testing. These hyperparameter settings and model choices are determined through empirical evaluation and existing literature on similar tasks.

## C Detailed Cognitive Distortion Detection Results

As shown in Table 8, we also provide some experimental results for C2D2-E. Similar to the results for C2D2, but the results for C2D2-E are generally higher. Apart from the impact of translation, we attribute this to the improved ability of the existing model to comprehend English, especially for today's LLM.

| Model | $F1$ | $Rec$ | $Pre$ |
|---|---|---|---|
| Bert | 0.72 | 0.74 | 0.71 |
| Roberta | **0.74** | **0.75** | **0.74** |
| ChatGPT (Zero-shot) | 0.45 | 0.42 | 0.54 |
| ChatGPT (Few-shot) | 0.56 | 0.58 | 0.55 |

Table 8: Performance of baseline models for the C2D2-E tasks. All metrics are calculated using macro-averaging.

## D Cognitive Distortions Analysis on eRisk-2018

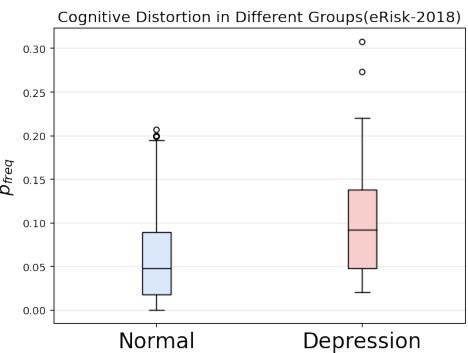

Figure 5: Box plot of the distribution of $p_{freq}$ across different groups on eRisk-2018.

As shown in Figure 5, we are presenting here the results obtained on the eRisk-2018 dataset, and without a doubt, the differences in cognitive distortions between the depression group and the normal group are once again evident. In this dataset, the proportion of normal users exhibiting cognitive distortions is higher than that in the Clpsych-2015 dataset.