# OpenReview forum: "C2D2 Dataset: A Resource for the Cognitive Distortion Analysis and Its Impact on Mental Health"
_EMNLP/2023/Conference — EMNLP 2023 Findings_

### Official Review · Reviewer_JxAP · 2023-07-24

**Soundness:** 3

**Excitement:**

3: Ambivalent: It has merits (e.g., it reports state-of-the-art results, the idea is nice), but there are key weaknesses (e.g., it describes incremental work), and it can significantly benefit from another round of revision. However, I won't object to accepting it if my co-reviewers champion it.

**Paper Topic And Main Contributions:**

This paper is about mental health analysis topic. The main contributions are: a publicly accessible Chinese Cognitive Distortion Dataset is constructed for the first time. The authors explore the association between cognitive distortions and various mental disorders on
social media. The authors attempt to incorporate information about users’ cognitive distortions into mental disorder detection models to illustrate the effectiveness of using cognitive distortion.

**Reasons To Accept:**

The C2D2 dataset is valuable for the mental health field. It is a publicly accessible resource within this domain that includes numerous texts recording individuals’ thoughts in various scenes with cognitive distortion. This dataset provides a profound connection between cognitive distortions and mental disorders. Addressing these distortions could potentially enhance mental health interventions. The authors find that integrating cognitive distortions as an additional feature can augment the performance of existing mental disorder detection models. In addition, the overall structure is well-organised.

**Reasons To Reject:**

The motivation "However, mental health issues in developing countries often receive limited attention" described for not annotating the dataset in English is inaccurate. English has a broader applicability, and the authors could also consider incorporating English annotations. While developing countries are essential to consider, it is also worthwhile to focus on datasets in languages other than Chinese.

For the test datasets, do the authors consider post-level model tests? Because the C2D2 is also the post-level dataset. Additionally, some other datasets could be added, like SWMH, T-SID, etc.
In Sec 6.2, the authors should train the RoBERTa-based model because of the performance shown in Table 5.
In Fig.4, do all posts with depression or PTSD have cognitive distortions? Add Non-sd labels?

There is no human expert performance on this dataset.

Many details need to be expressed clearly. What is the meaning of ppopularity in (4)?

The reasonable about "extracting posts that contain cognitive distortions" should be discussed.  Why not other methods?

If possible, it is suggested that the problem could be treated as a multi-label task when multiple cognitive distortions occur simultaneously.

**Reproducibility:**

4: Could mostly reproduce the results, but there may be some variation because of sample variance or minor variations in their interpretation of the protocol or method.

**Reviewer Confidence:**

4: Quite sure. I tried to check the important points carefully. It's unlikely, though conceivable, that I missed something that should affect my ratings.

---

> ### Author Rebuttal · Authors · 2023-08-29
>
> Thank you for investing the time to rigorously review our manuscript. We sincerely value your insights and commendations, which have played a vital role in elevating the quality of our work. Your feedback not only deepens our understanding but also offers a wider perspective, ensuring that our research stands up to the utmost academic scrutiny. We are genuinely appreciative of your constructive remarks that have considerably enriched our paper. Here are our detailed responses to your highlighted points:
>
> **1. Dataset Construction and Language Annotation:**
> We resonate with your concerns. Recognizing the universal relevance of English, we introduced the C2D2-E dataset. Despite the plethora of mental health datasets in English, there's a conspicuous lack of such datasets in the Chinese language. Our aspiration is that multilingual and cross-cultural datasets will spearhead the future endeavors in this sector.
>
> **2. Human Expert Performance on Our Dataset:**
> Your observation about the omission of human expert performance metrics is precise. For clarification, we subjected two psychology experts to a 500-entry sample from our dataset. Their evaluation reflected a macro-F1 score of 0.87 and kappa=0.76. Considering the inherent subjectivity of the task, we believe the performance of human experts is reasonable. These figures underline the potential for subsequent model refinement.
>
> **3. Clarifications on Experimental Setup:**
> - **Post-level Model Tests:** We believe that the mental disorder detection at the post-level might sometimes bypass the nuanced cognitive distortions. To illustrate, the statement "I must be a fool" exemplifies 'labeling', a cognitive distortion that might not be as prominently captured in some post-level annotations. Conversely, expressions such as "I have severe insomnia and feel nauseous" lean more towards symptomatic descriptions. It's also noteworthy that occasional cognitive distortions can be observed even in individuals without diagnosed mental disorders. Therefore, the connection between cognitive distortions and mental disorders should be established from social media through user's long-term social media history rather than a single post.
>
> >> User-level datasets contain this information because it includes the entire history of the patient, including daily thoughts. We acknowledge, however, that the efficacy of our datasets in informing post-level studies remains to be empirically validated. We greatly appreciate your insights, and they could very well shape the trajectory of our subsequent research efforts.
>
>
> - **Presence of Cognitive Distortions not in All Posts:** As shown in our example, It's crucial to highlight that not every post delineating mental disorders exemplifies cognitive distortions. A piece of content that seems immaterial on the surface might, from a psychologist's lens, be of utmost significance.
>
> - **Rationale Behind Extracting Posts with Cognitive Distortions:** Our methodology to detect mental disorders is premised on the identification of enduring cognitive distortion patterns in users. The most rudimentary strategy is to extract all potential posts containing cognitive distortions for modeling, serving as an indicator of long-term cognitive distortion. This approach harmonizes with our agenda of demonstrating the instrumental role of cognitive distortions in discerning mental disorders.
>
> - **Choice of Bert Model:** Your inquiry about the BERT vs. RoBERTa: We embraced BERT due to its universal endorsement as a standard in numerous computational endeavors. We aim to establish a benchmark for successive researchers in mental health detection. Your perspective makes sense; from a performance standpoint, RoBERTa should be the choice for analysis. However, our effort is inclined towards an analytical exploration and a foundational outlook. Therefore, we believe it sufficiently backs our findings and argument..
>
> **4. Issues with $p_{popularity}$:**
> We regret the oversight concerning the term $p_{popularity}$ in Equation (4). It was an inadvertent typo and should be read as $p_{freq}$, as delineated in Equation (1). We are thankful for your patience and kindly request any further clarifications on potential ambiguities.
>
> **5. Methodology and Multi-label Tasking:**
> The proposition of conceptualizing the problem as a multi-label task when numerous cognitive distortions emerge concurrently is an astute one. However, we believe that our current approach is also reasonable, as psychologists typically prioritize addressing the most evident cognitive distortions. At the same time, we need to take into account our annotation costs.
>
>
> Once again, we deeply appreciate your invaluable feedback and patience throughout this review process. Your insights offer us a clear direction to refine our paper. We believe that adjustments and additions will enhance the overall quality and depth of our work. We are confident we can address your concerns and further strengthen our contribution to the community. Thank you once again for your invaluable feedback.

---

### Official Review · Reviewer_jS9Q · 2023-08-02

**Soundness:** 4

**Excitement:**

4: Strong: This paper deepens the understanding of some phenomenon or lowers the barriers to an existing research direction.

**Paper Topic And Main Contributions:**

In this paper, the authors propose a cognitive distortion dataset in Chinese, consisting of 7,500 examples with paired scenes, thoughts, and cognitive distortion labels. For the dataset construction, the authors do not take the traditional approach of annotating online posts, but instead recruit volunteers and use interviews to further select qualified volunteers. The volunteers are instructed to compose thoughts based on the given scene, and also label the thoughts composed by other volunteers. The quality of the dataset is further evaluated by experts.

The authors conduct experiments using various baseline models including fully-supervised training, as well as few-shot and zero-shot using ChatGPT, demonstrating the challenge of this dataset. The author also conducts experiments and comprehensive analysis using the cognitive detection model on datasets from social media, showing the correlations between cognitive distortions and various mental health disorders.

**Questions For The Authors:**

Do you pay the volunteers and the experts? If so, how do you pay them? This needs to be specified in the Ethics section.

**Reasons To Accept:**

1. The proposed new dataset is very valuable, as currently, there is a lack of publically available datasets in the mental health domain. Such a dataset will greatly boost the research for NLP in assisting mental health treatment.
2. The proposed dataset is the first one for Chinese. This is especially valuable since, in China, the scarcity of mental health professionals and resources is even more severe. There is an increasing demand for building automatic assistance for mental health support.
3. The dataset shows good quality, verified by experts.
4. The experiments and analysis demonstrate the great potential of the vast scope of future research that can be extended based on this dataset.

**Reasons To Reject:**

1. No evaluation regarding the human expert performance on this dataset.
2. The dataset construction process involves asking the volunteers to "compose" distorted thoughts but not taking the thoughts from potential patients with mental disorders from online posts. Such a method, theoretically as a kind of proxy to the thoughts of real patients,  is further verified by experts, as stated in the paper. However, I'm wondering what the gap is between the thoughts from real patients and the composed thoughts.

**Reproducibility:**

4: Could mostly reproduce the results, but there may be some variation because of sample variance or minor variations in their interpretation of the protocol or method.

**Reviewer Confidence:**

4: Quite sure. I tried to check the important points carefully. It's unlikely, though conceivable, that I missed something that should affect my ratings.

---

> ### Author Rebuttal · Authors · 2023-08-29
>
> Thank you for the thoughtful feedback on our paper. We genuinely appreciate the time and effort you invested in reviewing our work. We are sincerely grateful for your recognition of the novelty of our work and your attention to the field of mental health. Your positive comments are very encouraging, and we are keen to address the concerns you raised.
> Please find below our responses to your queries:
>
> __1. Human expert performance on our dataset:__
>
> Your point on the absence of human expert performance metrics is valid. To clarify, we did evaluate the performance of two psychology experts on a 500-entry sample from our dataset, which yielded a macro-F1 score of 0.87. and kappa=0.76. Considering the inherent subjectivity of the task, we believe the performance of human experts is reasonable.
>
> __2.Gap between the thoughts from real patients and the composed thoughts:__
>
> We acknowledge and appreciate your concern. Our approach to having volunteers compose distorted thoughts ensured controlled and diverse data. To ensure the thoughts collected closely mirror reality, we screened volunteers based on scales. Those with personal experiences of cognitive distortions were included to add authenticity to the dataset. This process eliminated half of the potential volunteers. Furthermore, our review standards were stringent, involving multiple experts to ensure data reasonableness. Our goal has been to cooperate with experts and ensure data quality through rigorous processes.
>
> __3.Compensation for volunteers and experts:__
>
> __Do you pay the volunteers and the experts? If so, how do you pay them?__
>
> Yes, both volunteers and experts were compensated for their time and contribution. Before initiating the data collection, every participant had signed and confirmed our compensation agreement, which clearly defines the rights and responsibilities of both parties. Upon completion of the dataset annotation, we promptly compensated all participants as per the agreement and ensured that the funds were successfully transferred to their provided bank accounts.
>
> Once again, we deeply appreciate your invaluable feedback and patience throughout this review process. Your insights offer us a clear direction to refine our paper. We believe that adjustments and additions will enhance the overall quality and depth of our work. We are confident we can address your concerns and further strengthen our contribution to the community. Thank you once again for your invaluable feedback.

---

### Official Review · Reviewer_DcU8 · 2023-08-11

**Soundness:** 4

**Excitement:**

3: Ambivalent: It has merits (e.g., it reports state-of-the-art results, the idea is nice), but there are key weaknesses (e.g., it describes incremental work), and it can significantly benefit from another round of revision. However, I won't object to accepting it if my co-reviewers champion it.

**Missing References:**

The findings discussed in section 6.2 are similar to research on cognitive bias types in psychology although its novel being found on social media. Could support findings with research from psychology, psychiatry and the medical field.

**Paper Topic And Main Contributions:**

This paper discusses the creation of the Chinese Cognitive Distortion Dataset, a dataset based on Chinese tweets which were then translated into English. This dataset can be used to detect cognitive distortions (present or not) and the type of cognitive distortion (e.g., emotional reasoning, overgeneralization, etc) in text. It also examines its use in previously collected datasets with social media data from users with mental illnesses.

**Questions For The Authors:**

1) Will the C2D2 dataset be available in Chinese, English, or both?
2) What are the volunteer demographics?
3) Where/how were the volunteers recruited from?

**Reasons To Accept:**

Presents an interesting tool to be used to detect cognitive distortions and has an interesting method that could be replicated to re-create a cognitive bias dataset with another language (e.g., English, French, etc.,).

**Reasons To Reject:**

By collecting data from volunteers with prompts and then correcting the volunteer's answers (as discussed in section 3.4), the dataset may include the researcher's biases and be less generalizable or naturalistic. Unclear of potential biases in the demographics of the volunteers who generated the data. The biases and limitations of the dataset due to it being generated by volunteers and then corrected should be discussed. As well, since the authors use social media datasets it's unverifiable if the users are formally diagnosed, thus the findings discussed on 449 to 453 and 465 to 471 as well of the rest of the section should emphasize how these findings are based on self-disclosure and potential self-diagnosis.

**Reproducibility:**

4: Could mostly reproduce the results, but there may be some variation because of sample variance or minor variations in their interpretation of the protocol or method.

**Reviewer Confidence:**

4: Quite sure. I tried to check the important points carefully. It's unlikely, though conceivable, that I missed something that should affect my ratings.

**Typos Grammar Style And Presentation Improvements:**

Figure 2, in the notes missing spaces between the "&" in the first line

line 203, missing spaces between the "&".

Line 349, extra space before "Table"

---

> ### Author Rebuttal · Authors · 2023-08-29
>
> Thank you immensely for the detailed and constructive feedback on our paper. We truly appreciate the time and effort you invested in evaluating our work. It is heartening to know that you found our approach intriguing and see its potential for replication in other languages. Your endorsement is encouraging for us. Please find below our responses to your queries:
>
>
> __1.Availability of the Dataset:__
>
> **Will the C2D2 dataset be available in Chinese, English, or both?**
>
> Both Chinese and English are available. In point of fact, we intend to release our CD2D-E to facilitate wider recognition and replication of our research and findings. Specifically, the release of the CD2D-E aims to foster a broader understanding and replication of our research. While we want researchers to engage with the original C2D2 to appreciate the cultural nuances and complexities of mental well-being, we deemed the English C2D2-E necessary for global inclusivity.
>
> **2. Addressing Researcher Biases in the Dataset:**
>
> We acknowledge your concerns regarding potential researcher biases. It's pertinent to mention that our volunteer selection was thorough, leveraging social media platforms for initial registration followed by a meticulous selection process. This process was carefully calibrated to ensure gender and age balance, thereby mitigating biases. As delineated in our paper, each volunteer had firsthand experience with cognitive distortions and received specialized psychological training by us.
>
> Although the inherent intricacies of cognitive distortions did necessitate expert corrections, we ensured multiple layers of checks. Established standardized protocols, encompassing training, labeling, data inspection, and correction, were subjected to review by multiple specialists. This collaborative approach, involving multiple experts in discussions and corrections, was strategized to curtail the subjective inclinations of any single individual. We concede that absolute objectivity is challenging. However, our rigorous methodologies were designed to attenuate any potential biases to the maximum extent.
>
>
> __3.Perspective on Self-disclosure and Potential Self-diagnosis::__
>
> We concur that relying solely on self-disclosure poses inherent challenges, especially given the nuances of mental health disclosures on social platforms. But using datasets based on self-disclosure has become a common practice due to the inherent challenges in obtaining accurate mental health information on social media platforms. The publishers of the dataset we used say that they have vetted the credibility of the self-disclosed content, hence our decision to utilize it directly. Nevertheless, your emphasis on the tentative nature of such disclosures is well-taken.
>
> __4.Support from the Fields of Psychology, Psychiatry, and Medicine:__
>
> Thank you for highlighting the potential relevance of references from psychology, psychiatry, and medicine to our study. We believe that our findings in section 6.2 offer perspectives that traditional psychological research often struggles to access directly. Our analysis, grounded in objective data spanning hundreds of thousands of entries, provides insights from angles seldom explored in conventional psychological studies.
>
> __5.Addressing Typographical and Formatting Issues:__
>
> We deeply regret the oversight in spellings and format. We assure you of a thorough proofreading to ensure these are rectified.
>
> We are thankful for your comprehensive review and the clarity of your feedback. We are confident that our responses will elevate the overall quality and depth of our work. Your review has been pivotal in highlighting areas for enhancement, and we are optimistic about addressing your concerns to further solidify our contribution to the community.

---

### Official Review · Reviewer_8gtz · 2023-08-11

**Soundness:** 4

**Excitement:**

4: Strong: This paper deepens the understanding of some phenomenon or lowers the barriers to an existing research direction.

**Missing References:**

Not directly about cognitive distortions, but on the relation between the more general "cognitive styles" (including cognitive biases) and mental health disorders:

Uban, A. S., Chulvi, B., & Rosso, P. (2021). An emotion and cognitive based analysis of mental health disorders from social media data. Future Generation Computer Systems, 124, 480-494.

**Paper Topic And Main Contributions:**

The paper introduces a new dataset of examples of various cognitive distortions in the Chinese language. The authors then perform machine learning experiments to test whether cognitive distortions are possible to detect automatically from text, and also test their relation to mental disorders, based on some English datasets annotated for depression and PTSD.

The dataset is created through a complex and rigorous process: by first doing a selection for choosing the set of responders, then asking them to produce examples of cognitive distortions, which are then verified by experts for different criteria. The dataset is made public.

In the experiments, pretrained LLMs are used to detect cognitive distortions in the proposed dataset and in the English mental disorders dataset, using machine translation to make the datasets compatible (following two objectives: measuring the prevalence of cognitive distortions in mental health disorder data using the trained models, and trying to improve mental disorder detection using cognitive distortion information).

All the responders used for creating the dataset have agreed to participate in the study, and the ethical section contains a thorough discussion.

**Questions For The Authors:**

Will the English version of the dataset also be made publicly available? It might widen the scope of its usefulness in other studies.

**Reasons To Accept:**

- An original topic and the introduction of a new dataset, created through a very thorough process
- Interesting findings related to the specific cognitive distortions that occur in various mental health disorders (depression and PTSD)

**Reasons To Reject:**

- Some potential methodological problems or unclarities in the cross-lingual experiments. It would have been useful to see an analysis of the impact of translation on the dataset, since this might distort the annotations themselves - at least a reiteration of the cognitive distortion detection experiment but on the English version of the data

**Reproducibility:**

3: Could reproduce the results with some difficulty. The settings of parameters are underspecified or subjectively determined; the training/evaluation data are not widely available.

**Reviewer Confidence:**

3: Pretty sure, but there's a chance I missed something. Although I have a good feel for this area in general, I did not carefully check the paper's details, e.g., the math, experimental design, or novelty.

**Typos Grammar Style And Presentation Improvements:**

- line 72: will contributes -> contribute
- title of Section 4.2: "Related dataset" -> "Related datasets"?
- line 324: preferable not to start the paragraph with "And"
- lines 365-367: duplicate sentence (at least in meaning)?

---

> ### Author Rebuttal · Authors · 2023-08-29
>
> We sincerely thank you for your in-depth review of our paper. Your feedback is crucial to us, and we genuinely appreciate the time and effort you've taken to evaluate our work. We are pleased by your acknowledgment of the value of our exploration into cognitive distortions in mental health issues such as depression and PTSD. Conducting analyses traditionally deemed challenging by psychologists using computational techniques is indeed we want to show in our study.
>
> We genuinely value your constructive criticism and your emphasis on potential methodological considerations and opportunities for enhancement. Please find our responses to the points you raised below:
>
> 1.__Impact of English Translation:__
>
> We acknowledge the issue you've raised. We are indeed cognizant of this matter and due to space constraints, we included the results of the experiment involving English translations  in the appendix (Appendix B). Our experiment reveals that translation does indeed influence performance to some extent. However, the experiments suggest that the performance differences introduced by translation are within an acceptable range. Furthermore, we believe that the varying capabilities of modern models in handling Chinese and English also contribute to this aspect.
>
> __2. Availability of the Dataset:__
>
> __Will the English version of the dataset also be made publicly available?__
>
> Yes, we will release our C2D2-E. We are also grateful for your inquiry regarding the availability of the English version of our dataset. In point of fact, we intend to release our CD2D-E to facilitate wider recognition and replication of our research and findings. While our preference is for researchers to engage with the original Chinese version to foster a culturally diverse investigation into mental well-being, the release of C2D2-E is deemed necessary by us.
>
> __3.Remaining Issues:__
>
> We deeply regret the oversight in spellings and format. Regarding the syntactical and formatting matters you've highlighted, as well as any other oversights, we sincerely apologize. We are committed to rectifying these aspects. Furthermore, we acknowledge and are acting upon your suggestion regarding the addition of pertinent reference literature.
>
> Your commentary is immensely appreciated, and should you have any further suggestions, please do not hesitate to share them with us. We hope our response addresses your concerns and earns your further endorsement. We are truly thankful for your meticulous review of our paper. Your evaluation holds great significance for us, and we deeply value your insights.

---

### Meta-Review · Area_Chair_3hii · 2023-09-19

**Recommendation:** 3

**Metareview:**

This paper presents a Chinese corpus on cognitive distortions including testing some techniques for capturing cognitive distortions and some multi-lingual analysis for detecting depression and PTSD.

Overall strengths include a new dataset that is likely to be useful in the NLP for mental health space and an approach to create such data that could be useful in other languages while weaknesses include a lack of evaluation on the quality of human experts and a lack of integrating cognitive distortions, the main feature of the Chinese dataset, into the multilingual analyses.

---

### Decision · Program_Chairs · 2023-10-07

**Decision:**

Accept-Findings

**Comment:**

This paper presents a Chinese corpus on cognitive distortions including testing some techniques for capturing cognitive distortions and some multi-lingual analysis for detecting depression and PTSD.

Overall strengths include a new dataset that is likely to be useful in the NLP for mental health space and an approach to create such data that could be useful in other languages while weaknesses include a lack of evaluation on the quality of human experts and a lack of integrating cognitive distortions, the main feature of the Chinese dataset, into the multilingual analyses.